# Proinsulin degradation and presentation of a proinsulin B-chain autoantigen involves ER-associated protein degradation (ERAD)-enzyme UBE2G2

**Tom Cremer**[1,2], **Hanneke Hoelen**[1], **Michael L. van de Weijer**[3], **George M. Janssen**[4], **Ana I. Costa**[1], **Peter A. van Veelen**[4], **Robert Jan Lebbink**[1]*, **Emmanuel J. H. J. Wiertz**[1]*

**1** Department of Medical Microbiology, University Medical Center Utrecht, Utrecht, The Netherlands,
**2** Department of Functional Genomics, Center for Neurogenomics and Cognitive Research, VU University, Amsterdam, The Netherlands, **3** Sir William Dunn School of Pathology, University of Oxford, Oxford, United Kingdom, **4** Department of Proteomics and Metabolomics, Leiden University Medical Center, Leiden, The Netherlands

* E.Wiertz@umcutrecht.nl (EJHJW); r.j.lebbink-2@umcutrecht.nl (RJL)

## Abstract

Type 1 diabetes (T1D) is characterized by HLA class I-mediated presentation of autoantigens on the surface of pancreatic β-cells. Recognition of these autoantigens by CD8+ T cells results in the destruction of pancreatic β-cells and, consequently, insulin deficiency. Most epitopes presented at the surface of β-cells derive from the insulin precursor molecule proinsulin. The intracellular processing pathway(s) involved in the generation of these peptides are poorly defined. In this study, we show that a proinsulin B-chain antigen (PPI$_{B5-14}$) originates from proinsulin molecules that are processed by ER-associated protein degradation (ERAD) and thus originate from ER-resident proteins. Furthermore, screening genes encoding for E2 ubiquitin conjugating enzymes, we identified UBE2G2 to be involved in proinsulin degradation and subsequent presentation of the PPI$_{B10-18}$ autoantigen. These insights into the pathway involved in the generation of insulin-derived peptides emphasize the importance of proinsulin processing in the ER to T1D pathogenesis and identify novel targets for future T1D therapies.

## Introduction

Presentation of autoantigens in the context of HLA class I molecules on pancreatic β-cells plays an important role in Type 1 Diabetes (T1D) pathogenesis. These autoantigens are recognized by CD8+ T cells, present in pancreatic islands of T1D patients [1]. Autoreactive T cells destroy a significant part of the insulin-producing β-cell population, which results in severe insulin deficiency (reviewed in [2]). HLA-A*02:01-restricted CD8+ T cells have been implicated in the formation of pre-diabetic β-lesions. These T cells also accelerate the onset of T1D in HLA-A*02:01-transgenic NOD mice. Furthermore, the presence of this T cell population is associated with the development of T1D in high-risk class II HLA-DR3 and HLA-DR4 carriers

**Data Availability Statement:** Relevant data are within the manuscript and its Supporting Information files. Mass spectrometry spectra are

available from the ProteomeXchange consortium PRIDE database (identifier PXD049996; link: https://proteomecentral.proteomexchange.org/cgi/GetDataset?ID=PXD049996).

**Funding:** This research has been funded by the "Diabetes Foundation Expert Center Beta Cell Protection" funded by the Dutch Diabetes Foundation (DFN 2008.04.001), to HH and EW (http://www.diabetesfonds.nl/). This work was also financially supported by the Dr. Valliant Foundation, to HH and EW (http://www.lvc-online.nl/dr-c-j-vaillantfonds). The funders had no role in study design, data collection and analysis, decision to publish, or preparation of the manuscript.

**Competing interests:** The authors have declared that no competing interests exist.

[3, 4]. Most epitopes recognized by autoreactive CD8[+] T cells originate from insulin [5–7]. Evading immune responses evoked by proinsulin antigens by deleting insulin genes has been shown to prevent diabetes in the NOD mouse, emphasizing the importance of this T cell subset as a potential therapeutic target [6, 8].

Preproinsulin (PPI) molecules are both co- and post-translationally translocated into the ER, after which their signal peptide is cleaved off. The resulting proinsulin (PI) molecules subsequently mature, starting with the correct pairing of six cysteine residues to form three evolutionarily conserved disulfide bonds [9–11]. Most proinsulin molecules oligomerize and pass the Golgi to be sorted into secretory granules, where, following C-peptide cleavage, they achieve their mature form and await secretion upon glucose stimulation [10]. At the same time, excess or misfolded proinsulin molecules are removed from the ER by a quality control mechanism known as ER-associated protein degradation (ERAD) [12–14]. Erroneous proinsulin molecules are (partially) unfolded or disaggregated and prepared for ERAD by ER-resident reductases and chaperones including Grp170, PDIA, PDIA6, Calnexin and Calreticulin [13, 15–17]. Over one hundred ERAD components have been identified so far, and their complex cooperation is mostly centered around membrane-bound E3 ubiquitin ligases, which ubiquitinate ERAD substrates during dislocation across the ER membrane into the cytosol (reviewed in [18]). Upon dislocation of glycosylated substrates, asparagine-bound glycans are removed by the cytosolic enzyme *N*-glycanase [19] (illustrated in Fig 1A), resulting in deamidation of an asparagine (N) residue to aspartate (D) (Fig 1B). The process of attaching ubiquitin to a protein involves three steps, the first of which is activation of ubiquitin by the E1 enzyme in preparation of ubiquitin for further transfer. Subsequently, ubiquitin conjugating enzymes, or E2s, can receive activated ubiquitin from the E1. Finally, ubiquitin ligases, or E3s, select substrates for ubiquitination and facilitate transfer of one or more ubiquitin molecules to an acceptor residue [18].

More than a dozen ER-specific E3 enzymes have been identified thus far. Among these, HRD1 was found to be specifically involved in turnover of proinsulin, operating in an ERAD complex that contains Derlin-2, SEL1L and the AAA-ATPase p97 [12, 13]. P97 extracts ubiquitinated ERAD substrates from the ER membrane and shuttles them to the proteasome for degradation. If proteolysis of proteasomal substrates is incomplete, this process liberates short peptides suitable for re-import into the ER, the majority of which depends on Transporter associated with Antigen Presentation (TAP). In the ER lumen, peptides may be loaded onto HLA class I molecules and subsequently presented to CD8[+] T cells (Fig 1A). Several E2 conjugating enzymes have been implicated in ERAD, including UBE2G2, UBE2J1 and UBE2J2, with the former two reported to work in conjunction with HRD1 [20–22]. The identity of the E2 enzyme that is involved in degradation of proinsulin has remained elusive so far.

In this study, we show that a B-chain proinsulin antigen presented by HLA-A*02:01 derives from ER-resident proinsulin molecules dislocated to the cytosol during ERAD. Additionally, we use CRISPR/Cas9-mediated knockout of HRD1 to demonstrate its involvement in degradation of proinsulin and identify UBE2G2 as an E2 conjugating enzyme for ERAD-mediated proinsulin quality control and subsequent antigen presentation of a clinically relevant B-chain epitope (PPI$_{B10-18}$). These results shed new light on the route of proinsulin antigen processing and implicate ERAD's molecular machinery in this process.

## Materials & methods

### Antibodies and chemicals

Western blot analysis was performed with the following antibodies: H86 for proinsulin (Santa Cruz, cat. # sc-9168), C4 for beta-actin (Millipore, lot. #LV1728681), H68.4 for transferrin-receptor (Invitrogen, cat. # 13–6800), mouse HRD1 (Cell Signaling, #12925), D8Z4G for

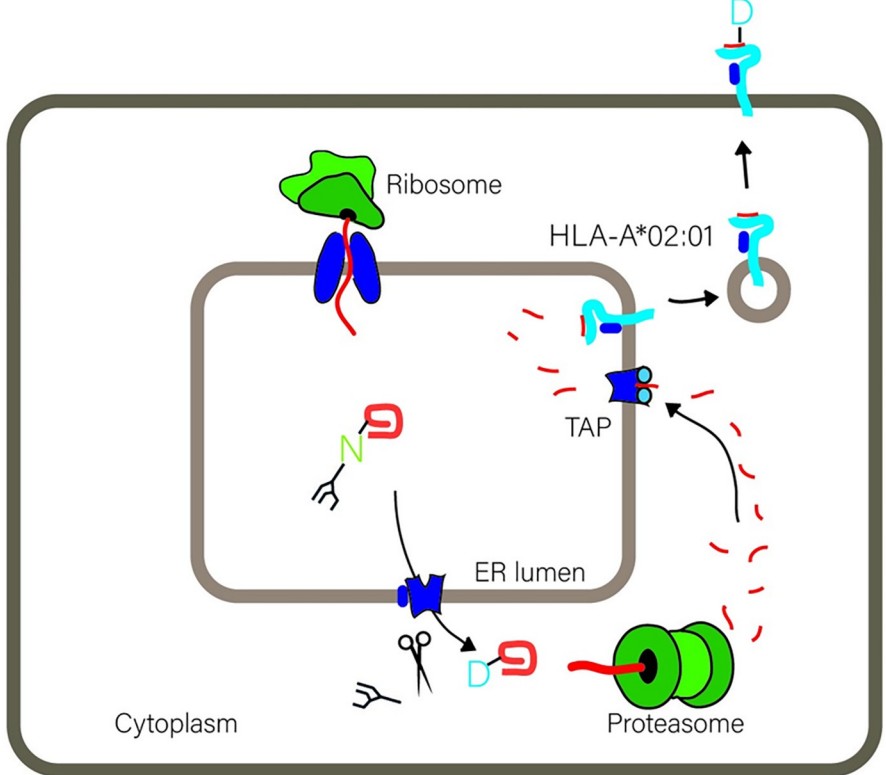

## A  Cell surface presentation of deamidated peptide

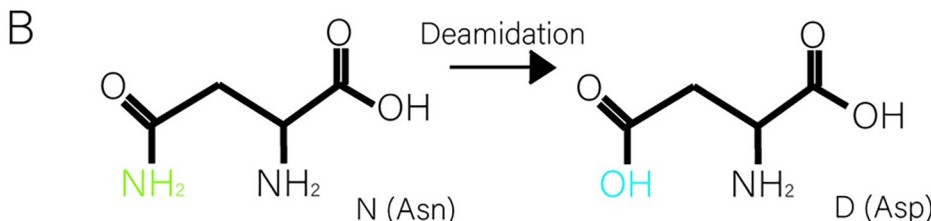

**Fig 1. Dislocation of ERAD substrates over the ER membrane is followed by deamidation. A.** In this study, we introduce a glycosylation consensus sequence in proinsulin to follow epitope trafficking. Schematic representation of the degradation route of PI-C31N. PI-C31N molecules are co-translationally translocated into the ER where N-linked glycosylation takes place. Dislocation of Δss-PPI-C31N molecules is accompanied by the removal of the N-linked glycan, leaving the protein deamidated. Dislocated insulin molecules are then targeted to the proteasome and the resulting proinsulin epitopes are imported into the ER by TAP, loaded onto HLA class I molecules and presented at the cell surface. **B**. Deamidation of an asparagine residue results in an aspartate residue.

UBE2g2 (Cell Signaling, #63182), and HCA2 and W6/32 against HLA. Secondary antibodies used were goat α-mouse IgG(H+L)-HRP (no. 170–6516, Bio-Rad) and goat α-rabbit IgG(H+L)-HRP (no. 4030–05, Southern Biotech). Cycloheximide was from Enzo Life Sciences.

## Constructs

PPI and its glycosylation mutants and PPI-GFP were expressed from a pSico-based lentiviral plasmid [23] containing an EF1α-promotor and Zeocin and ampicillin resistance genes.

CRISPR-Cas9 plasmids and E3- and E2-enzyme specific gRNAs used in the E2 screen have been described earlier [22, 23] and gRNA sequences can be found in S1 Table.

## Cell culture

K562 cells stably expressing proinsulin [12] and HEK293T cells for virus production were cultured in IMDM (Lonza). All media were supplemented with 10% FCS, 100U/mL penicillin, 100U/mL streptomycin and 2mM GlutaMAX. Cell lines were all maintained in humidified incubators at 37˚C and 5% $CO_2$.

## Lentiviral transductions

Replication-deficient recombinant lentiviruses were produced via Mirus-mediated co-transfection of HEK293T cells with a pSico-based lentiviral vectors encoding the gene of interest, together with pCMV-VSVG, pMDLg-RRE, and pRSV-REV [12]. After 48–52 hours, supernatant was harvested. Transductions were performed by 90-minute spin-oculation at 2000 RPM. Cells were allowed to recover for 48 hours after which they were selected with the appropriate antibiotic or sorted for fluorescent protein expression by FACS.

## Generation of clonal knockout cell lines with CRISPR/Cas9-mediated genome editing

K562 cells expressing HLA-A*02:01 and PPI were transduced with a lentiviral CRISPR/Cas9 vector, in which a single lentiviral plasmid encodes Cas9, puroR and a gRNA sequence (HRD1-1: GTGATGGGCAAGGTGTTCTT; HRD1-2; GCGGCCAGCCTGGCGCTGAC; HRD1-3: GGCCAGGGCAATGTTCCGCA; UBE2G2-1: GGAGAAGATCCTGCTGTCGG; UBE2G2-2: GGACTTAACGGGTAATCAAG; UBE2G2-3: GACTTAACGGGTAATCAAGT). After puromycin selection (0,2mg/mL), single clones were created by limiting dilution and knock-out status was confirmed by Western blot and genomic target site sequencing. Rescue cell lines were obtained by transduction with guide-resistant cDNA expressed from a lentiviral plasmid which also encodes an mAmetrin cassette, followed by sorting of mAmetrin positive cells with a FACSAriaII (BD).

## Arrayed flow cytometry screen

For the arrayed CRISPR-Cas9-mediated knockout screen, polyclonal knockout populations were analyzed for GFP (FITC) signal 9 days post-infection. Samples were fixed in 0,1% PFA before flow cytometric analysis by a FACSCantoII cytometer (BD). eGFP signal in live (FSC/SSC) gated cells was quantified to assess total PPI-GFP levels. Data of two technical replicates was averaged and normalized to control gRNA samples. Data analysis was performed using FlowJo software (TreeStar), Excel (Microsoft), and Graphpad PRISM 6 software (Graphpad).

## Western blot

1% Triton cell lysates were treated with 50mM DTT and 100mM NEM before loading on a 12% Nu-PAGE gel (Invitrogen). Separated proteins were blotted on PVDF membranes (Millipore), blocked in 5% ELK in TBS (150mM NaCl, 3μM KCl, 25 mM Tris-HCl pH 7.5) + 5% ELK and stained with primary and secondary antibodies in TBS + 0,1% Tween-20. Detection was performed using ECL substrate solutions (Pierce) and an Imagequant LAS4000 luminescent image analyzer (GE Healthcare). Membrane scans were analyzed using ImageJ software. Statistical significance was reported from three independent, actin-normalized experiments.

### RT-PCR

mRNA transcripts were isolated using a RNeasy kit (Qiagen) and converted to cDNA using random primers and M-MuLV RT-PCR enzyme (NEB). Proinsulin-specific primers (GTGAACCAGCACCTGTGC Fw and CGGGTCTTGGGTGTGTAGAAG Rv) were used to amplify cDNA in a standard PCR reaction, which was analyzed on a 2% agarose gel.

### Peptide elution & mass-spectrometry

Approximately $1x10^9$ cells from PPI (or variant) expressing cells were lysed in 10 ml lysis buffer (50 mM Tris-Cl pH 8.0, 150 mM NaCl, 5 mM EDTA, 0.5% Zwittergent 3–12 (N-dode-cyl-N,N-dimethyl-3-ammonio-1-propanesulfonate) and protease inhibitor (Complete, Roche Applied Science)) for 2 h at 0˚C (ref A). Lysates were successively centrifuged for 10 min at $2500 \times g$ and for 45 min at 31,000 x g to remove nuclei and other insoluble material, respectively. Lysates were passed through a 100 µl CL-4B Sepharose column to preclear the lysate and subsequently passed through a 100 µl column containing 250 µg pan class I (W6/32) IgG coupled to protein A Sepharose (ref A). The W6/32 column was subsequently washed with lysis buffer, low salt buffer (20 mM Tris-Cl pH 8.0, 120 mM NaCl), high salt buffer (20 mM Tris-Cl pH 8.0, 1 M NaCl), and finally low salt buffer. HLA α chain, β2m and peptides were eluted with 10% acetic acid, diluted with 0.1% TFA and purified by SPE (Oasis HLB, Waters) by sequential elution with 20% and 30% acetonitrile in 0.1% TFA to remove HLA protein chains. Peptides were lyophilized, dissolved in 95/3/0.1 v/v/v water/acetonitrile/formic acid and subsequently analyzed by online C18 nanoHPLC MS/MS with a system consisting of an Easy nLC 1200 gradient HPLC system (Thermo, Bremen, Germany), and a LUMOS mass spectrometer (Thermo). Fractions were injected onto a homemade precolumn (100 µm × 15 mm; Reprosil-Pur C18-AQ 3 µm, Dr. Maisch, Ammerbuch, Germany) and eluted via a home-made analytical nano-HPLC column (30 cm × 50 µm; Reprosil-Pur C18-AQ 3 um). The gradi-ent was run from 2% to 36% solvent B (20/80/0.1 water/acetonitrile/formic acid (FA) v/v) in 120 min. The nano-HPLC column was drawn to a tip of ∼5 µm and acted as the electrospray needle of the MS source. The LUMOS mass spectrometer was operated in data-dependent MS/MS mode for a cycle time of 3 seconds, with a HCD collision energy at 32 V and recording of the MS2 spectrum in the orbitrap. In the master scan (MS1) the resolution was 60,000, the scan range 400–1400, at an AGC target of 400,000 @maximum fill time of 50 ms. Dynamic exclusion after n = 1 with exclusion duration of 20 s. Charge states 1–3 were included. For MS2 precursors were isolated with the quadrupole with an isolation width of 1.2 Da. First mass was set to 110 Da. The MS2 scan resolution was 30,000 with an AGC target of 50,000 @maximum fill time of 100 ms. In a post-analysis process, raw data were first converted to peak lists using Proteome Discoverer version 2.5 (Thermo Electron), and then submitted to the minimal Uniprot Homo sapiens database (20596 entries), using Mascot v. 2.2.07 (www.matrixscience.com) for protein identification. Mascot searches were with 10 ppm and 0.02 Da deviation for precursor and fragment mass, respectively, and no enzyme was specified. Methi-onine oxidation was set as a variable modification. Peptides with MASCOT scores <35 were generally discarded.

## Results

### HLA-A*02:01 restricted presentation of the PPI$_{B5-14}$ epitope involves dislocation of proinsulin over the ER membrane

First, we investigated whether presentation of proinsulin-derived epitopes by HLA class I involves dislocation of the ER membrane. For this, we used a K562 cell line stably expressing

HLA-A*02:01 [12], which presents proinsulin peptides according to the β-cell's proinsulin-derived HLA-A2 peptidome [24]. As mentioned before, N-glycosylated ERAD substrates are deglycosylated in the cytoplasm. The intrinsic deamidation of the N-glycosylated asparagine allows us to investigate whether a proinsulin B-chain epitope (PPI$_{B5-14}$: HLCGSHLVEA) travels from the ER back into the cytosol. To this end, we designed a PPI construct which encodes an asparagine residue at position 31, thereby creating an N-glycosylated consensus motif (PPI-C31N, Fig 2A). This motif is present in the PPI$_{B5-14}$ peptide that is displayed on HLA-A*02:01 on the cell surface of our established surrogate B cell system. Involvement of ERAD in presentation of this proinsulin mutant would thus result in presentation of a peptide in which the asparagine has been converted into an aspartate residue. Three mutants were

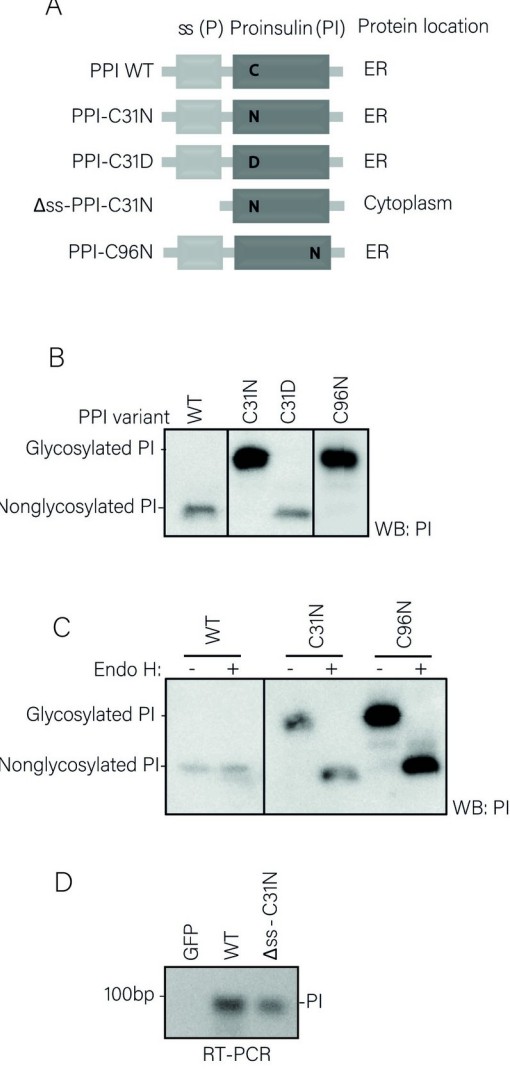

**Fig 2. Expression of proinsulin mutants in K562 cells. A.** Schematic overview of constructs that were used in the experiments in this figure and mass-spectrometry analysis in Fig 6. **B.** Expression of mutant proinsulin constructs in cells detected by Western blot. Shorter exposure for C96N mutant is shown because of high protein levels. Fragments of the original image were spliced together (see S4 Fig for raw scans) **C.** Cell lysates were treated with Endo H before Western blot analysis of the removal of N-linked sugar groups from glycosylation proinsulin mutants. Fragments of the original image were spliced together (see S4 Fig for raw scans) **D.** Detection of expression of Δss-PPI-C31N mRNA species by RT-PCR.

used as controls. First, a C31D mutant, which carries an aspartate residue from the beginning. Secondly, a C31N mutant lacking the signal sequence (Δss-C31N), which does not enter the ER, is not glycosylated, and not subjected to dislocation, therefore subsequently leaving the asparagine residue for presentation at the cell surface. Third, a C96N mutant, which harbors a glycosylation motif within a non-presented sequence and, like C31N, lacks a cysteine residue involved in proinsulin folding. We stably expressed the different PPI constructs in K562 HLA-A*02:01 cells (Fig 2B). Importantly, C31N and C96N mutants were properly glycosylated since both mutant proteins were sensitive to Endo H treatment (Fig 2C). The Δss-C31N mutant is degraded very rapidly [25]. Therefore, we validated ectopic expression of Δss-C31N mRNA transcripts via RT-PCR (Fig 2D). These results indicate that all cell lines properly express and process mutant proinsulin molecules. To assess HLA content from these cells, HLA class I molecules were affinity purified and the peptide ligandome was acid-eluted and analyzed by mass spectrometry. Peptides containing the aspartate (D) residue could be detected only on HLA molecules from cells expressing either the C31D or the C31N mutant, and not on cells expressing *wt* PI, Δss-C31N or the C96N mutant (Fig 3A, S1 Fig). Importantly, disruption of a disulfide bond in change of a glycosylation motif did not affect presentation of $PPI_{B5-14}$ peptides (Fig 3A, C96N cells). The $N^{31}$-containing peptide was predicted to bind to HLA as well as wild-type proinsulin-derived peptides (0,348 vs 0,344 for wt vs C31N,

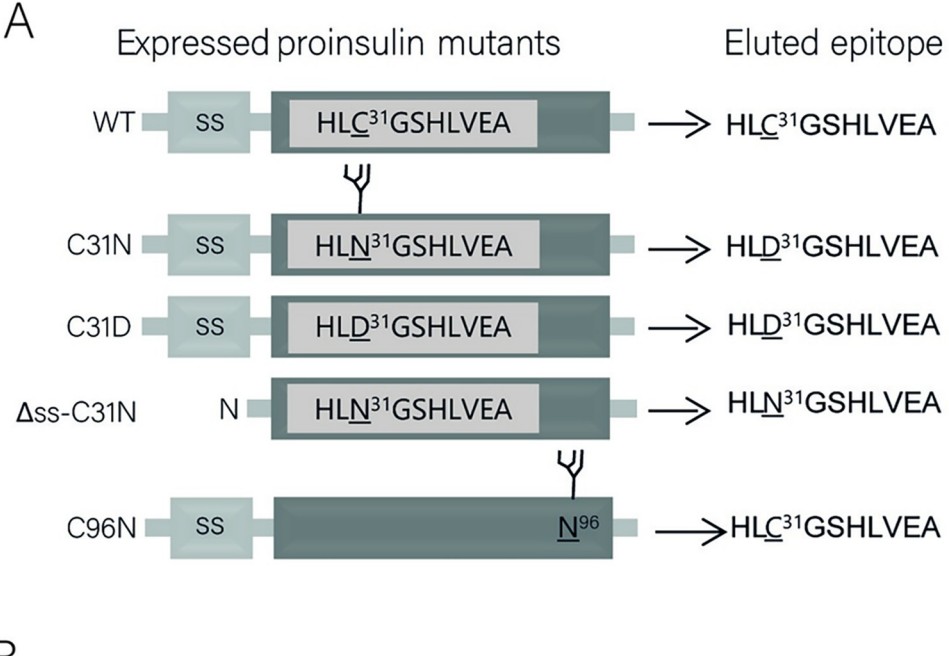

**B**

| Peptide | HLA-A02:01 predicted affinity (Kd, nM |
|---|---|
| HLCGSHLVEA | 1158 |
| HLNGSHLVEA | 1214 |
| HLDGSHLVEA | 590 |

**Fig 3. Presentation of the $PPI_{B5-14}$ epitope requires dislocation of proinsulin over the ER membrane. A.** Representation of mass-spectrometry results of HLA-bound peptides that were eluded from affinity-purified MHC molecules derived from K562 HLA-A2 cells expressing the indicated mutant proinsulin constructs (see S3 Fig for spectra of eluted and synthetic peptides) **B.** Affinity of peptides that result from processing of mutant proinsulin constructs for HLA-A*02:01 molecules as predicted with NetMHC 3.4 (http://www.cbs.dtu.dk/services/NetMHC/).

respectively, see Fig 3B) (*Net*MHC, [26]), and this was confirmed by mass-spectrometry analysis (Fig 3A, Δss-C31N cells, S1 Fig). Altogether, these results identify proinsulin molecules that are dislocated over the ER membrane as a source presented proinsulin B-chain (PPI$_{B5-14}$) peptides.

## HRD1 catalytic activity is involved in proinsulin degradation

Since dislocation over the ER membrane relies on ERAD machinery, we next set out to identify ERAD factors involved in proinsulin degradation. To confirm the involvement of the E3 ubiquitin ligase HRD1 in this process [12, 13], we completely depleted cells of HRD1 by CRISPR/Cas9-mediated genome editing. K562 cells expressing HLA-A*02:01 and wild-type PPI were transduced with guideRNAs (gRNAs) targeting the region encoding for the 5' end of the HRD1 gene. Each of the gRNAs tested reduced HRD1 protein levels in a polyclonal cell population (S2A Fig). Clonal cell lines were established and analyzed for Cas9-mediated indel formation. Out-of-frame indels, which result in a frameshift that may lead to a premature stop codon, were present in three selected clonal cell lines (Fig 4A and S2B Fig). Loss of HRD1 expression was confirmed by Western blotting (Fig 4B and S2C Fig). Next, we re-transduced selected K.O. cell lines to express either wild-type HRD1 or a mutant in which the first cysteine residue within the HRD1 RING domain was changed to an alanine, rendering the mutant catalytically inactive (C1A). HRD1 or HRD1 C1A expression could be detected in these cell lines at similar protein levels (Fig 4B and S2C Fig). Importantly, compared to KO cells, re-expression of HRD1 WT, but not HRD1-C1A, decreased PI steady-state levels. Furthermore, proinsulin is relatively stable in HRD1 KO cells, as observed in a CHX chase assay (Fig 4C). Reintroduction of HRD1 WT, but not HRD1-C1A, enhanced proinsulin degradation (Fig 4C and 4D), indicating that the first catalytic cysteine residue within HRD1's RING domain is required for proinsulin degradation. These results confirm earlier studies and indicate that HRD1 catalytic activity is required for efficient degradation of proinsulin molecules.

We next investigated whether HRD1-mediated proinsulin degradation is required for cell surface presentation of proinsulin-derived peptides. Therefore, we subjected the HLA-bound peptidome from our panel of K562-HLA-A*02:01 cells to mass spectrometry and analyzed the relative abundance of cell-surface presented peptides as a function of HRD1. Presentation of two proinsulin-derived peptides (PPI$_{A15-24}$: ALWGPDPAAA and PPI$_{B10-18}$: HLVEALYLV), but also that of other peptides was highly affected by presence or activity of HRD1 (Fig 4E). Interestingly, HRD1 knockout or reintroduction of HRD1-C1A generally inhibited the abundance of peptides on HLA-A*02:01, while expression of WT HRD1 restored antigen presentation (Fig 4E). Overall, our results show that HRD1 activity is required for proinsulin degradation and suggest that it has a general role in HLA-A*02:01-mediated antigen presentation.

## UBE2G2 activity controls proinsulin degradation and presentation of a proinsulin-derived autoantigen

After confirming HRD1 involvement in PI degradation, we set out to identify the E2-enzyme that acts in concert with HRD1 to target proinsulin molecules to the proteasome. We used an arrayed CRISPR-Cas9 library comprised of 120 gRNAs to target 40 mammalian E2 enzymes (3 gRNAs/gene) [22]. K562 cells stably expressing a proinsulin-GFP fusion protein (PI-GFP) were transduced with E2-specific CRISPR vectors (S1 Table). Steady state PI-GFP levels were assessed in the gRNA-positive cell population by flow cytometry at 9 days post-infection. Comparable PI-GFP levels were detected in most knock-out cells, except in cells that expressed gRNAs targeting the ubiquitin conjugating enzyme UBE2G2 (Fig 5A, S2 Table). A substantial

A

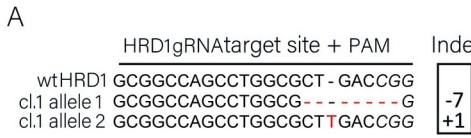

B

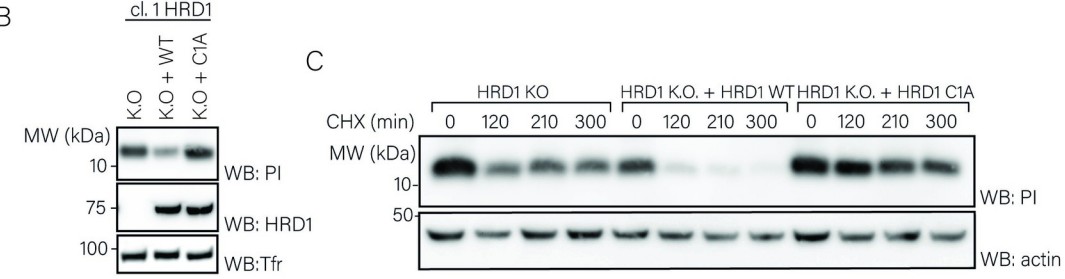

C

D

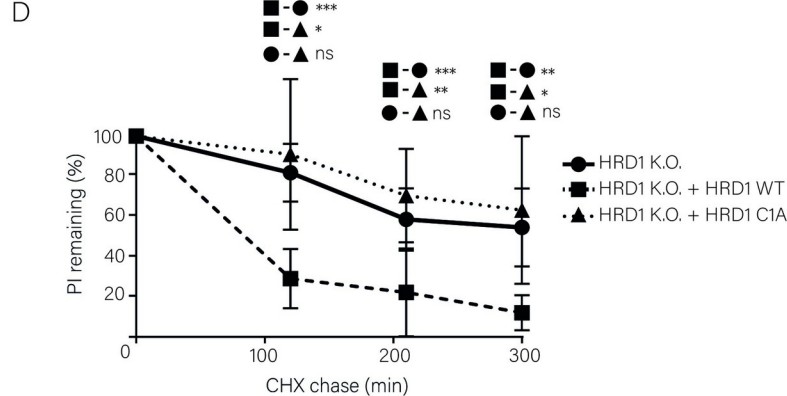

E

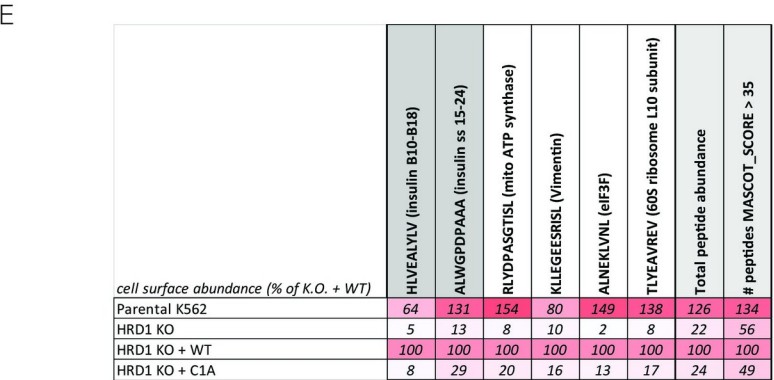

**Fig 4. HRD1 catalytic activity is involved in proinsulin degradation. A.** K562 cells stably expressing HLA-A*02:01 and PPI were transduced with a lentiviral CRISPR-Cas9 vector containing a gRNA sequence directed against the 5' region of the HRD1 gene and selected with puromycin. Monoclonal knockouts were generated by limited dilution and analyzed for HRD1 expression. Genomic DNA was isolated and sequenced for the presence of insertions of deletions (indels) within the gRNA target region. Both HRD1 alleles were aligned to a reference sequence (NCBI gene entry 84447). **B.** K562 cells from (A) were re-transduced with an empty cDNA vector (KO) or a cDNA vector encoding HRD1 (WT), or a catalytically inactive mutant (C1A). Cells were sorted based on mAmetrin expression to obtain pure populations. Next, Expression of HRD1 and proinsulin was assessed in corresponding cell lysates by immunoblotting. Human transferrin receptor was used as a loading control. **C.** K562 cells from (B) were treated with 200ug/mL CHX for the indicated times, followed by WB analysis of total proinsulin levels. A representative blot from three independent experiments is shown here. **D**. Results from 5 independent experiments of (C) were quantified, corrected for actin levels and normalized to t = 0 (Error bars = SD, paired Students' T-test; *p<0,05; **

p<0,01; *** p<0,001, ns = non-significant). Data points are shown in S2 Table. **E.** The HLA-eluted peptidome from (rescued) HRD1 K.O. clones from (B) or parental cells was analyzed by mass spectrometry (N = 1) and quantified from MASCOT output. Abundance of specific peptides is shown as a percentage of their presentation in HRD1 K.O. + HRD1 WT rescued cells, in a white-red gradient-coded table. For raw data, see S3 Table.

increase of 40–50% was observed in all three cases, suggesting that UBE2G2 is an important factor in degradation of proinsulin. Notably, PI-GFP was not rescued in cells expressing gRNAs against UBE2G1, UBE2G2's closest homologue, pointing at the specificity of the observed phenotype (Fig 5B).

Next, we set out to validate these results in clonal UBE2G2 knockout cells expressing untagged proinsulin. Using the gRNA that showed the most efficient depletion of UBE2G2 in the polyclonal knockout population, gRNA#1 (S3A Fig), three clonal cell lines were created and analyzed for Cas9-mediated genome editing. We observed out-of-frame indels in selected clones (Fig 6A and S3B Fig), corresponding to the loss of UBE2G2 expression (Fig 6B and S3C Fig). UBE2G2 knockout cells showed increased steady state proinsulin levels (Fig 6B). To confirm that this phenotype depends on loss of UBE2G2 activity, we added back gRNA-resistant cDNAs encoding wild-type HA-tagged UBE2G2 (WT) or a catalytically inactive (C89S) mutant of UBE2G2. HA-UBE2G2 expression could be detected in these cells at similar levels,

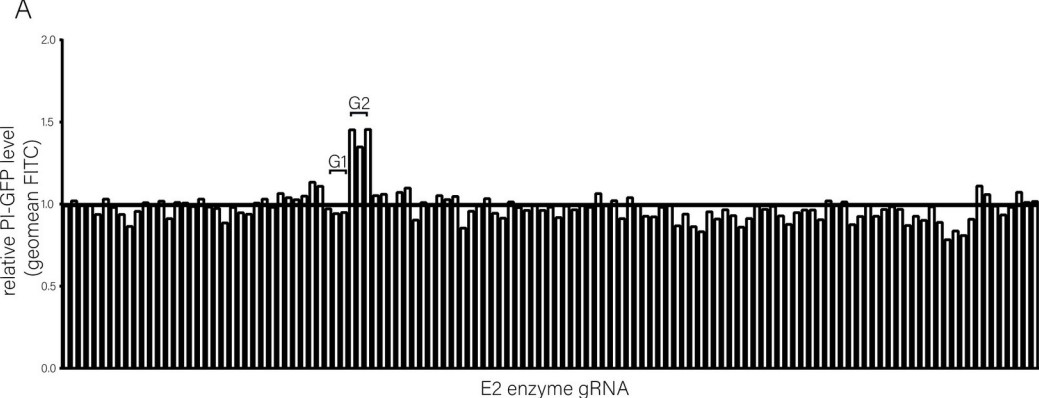

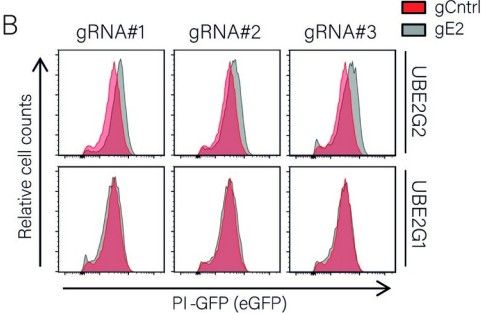

**Fig 5. An E2-specific CRISPR knockout screen identifies UBE2G2 to be involved in PI-GFP degradation. A.** An arrayed CRISPR library targeting every known human E2-conjugating enzyme with three different gRNAs per gene [22] was transduced into K562 cells expressing PPI-GFP. Cells were selected with puromycin and GFP levels were assessed by flow cytometry 9 days post-infection. Results were quantified and compared to GFP levels in empty vector-expressing cells. For sample IDs, see S1 Table. **B.** Histograms of cells transduced with gRNAs targeting UBE2g2 or UBE2g1 (grey) or an empty vector control (red) from the screen shown in (A). PI-GFP levels were evaluated using flow cytometry. Raw values and calculations are shown in S2 Table.

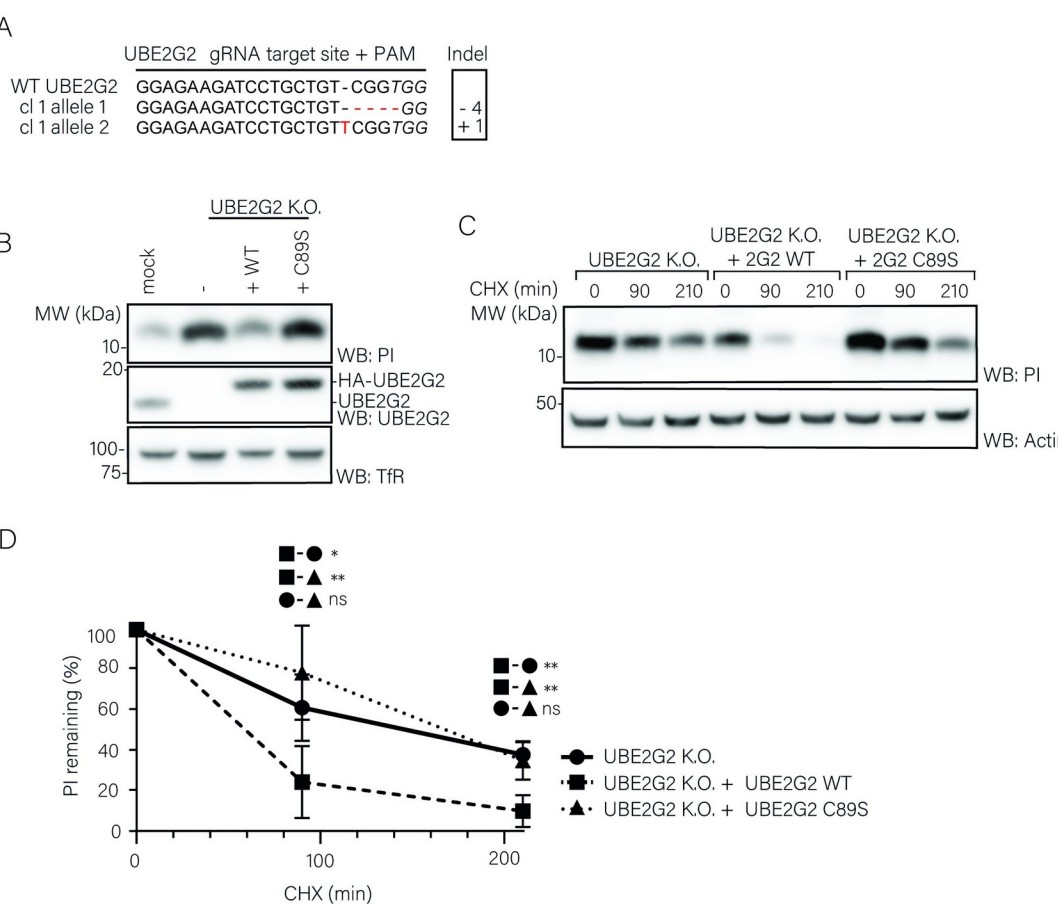

**Fig 6. UBE2G2 catalytic activity is involved in proinsulin degradation. A.** K562 cells stably expressing HLA-A*02:01 and PPI were transduced with a CRISPR-Cas9 vector containing a gRNA sequence directed against the N-terminal region of the UBE2g2 gene and selected with puromycin. Monoclonal knockouts were generated by limited dilution and analyzed for UBE2g2 expression. Genomic DNA was isolated and sequenced for the presence of insertions of deletions (indels) within the gRNA target region. Both UBE2g2 alleles were aligned to a reference sequence (NCBI gene entry 7327). **B.** K562 cells from (C) were retransduced with an empty cDNA vector (KO) or a gRNA-resistant cDNA vector encoding *wt* HA-UBE2G2 (WT), or a catalytically inactive mutant HA-UBE2G2 (C89S). Cells were sorted for mAmetrin to enrich for gRNA-positive cells. Next, cell lysates were analyzed by WB for expression of UBE2G2 and proinsulin. Human transferrin receptor (TfR) was used as a loading control. **C.** K562 cell lines from (B) were treated with 200ug/mL CHX for the indicated times, followed by WB analysis of total proinsulin levels. A representative blot from three independent experiments is shown. **D.** Results from 6 independent

experiments of (C) were quantified, corrected for actin levels and normalized to t = 0 (Error bars = SD, paired Students' T-test; *p<0,05; **p<0,005, ns = non-significant) Data points are shown in S2 Table. **E.** The HLA-eluted peptidome from (rescued) UBE2G2 K.O. clones from (B) and parental cells was analyzed by mass spectrometry (N = 1) and quantified from MASCOT output. Abundance of specific peptides and total (>35 ion scored) peptides is shown as a percentage of their presentation in UBE2G2 K.O. + WT UBE2G2 cells in a white-red gradient-coded table. See S3 Table.

and PI degradation in UBE2G2 KO cells could be restored by expression of WT UBE2G2, but not by expression of C89S UBE2G2, in multiple isogenic settings (Fig 6B, S3C Fig). Importantly, as shown by a CHX chase assay (Fig 6C and 6D), this increase in protein level is the result of a decreased proinsulin degradation rate that specifically depends on the UBE2G2 C89 catalytic residue. These results clearly indicate that UBE2G2 and its ubiquitin conjugating activity are involved in proinsulin degradation.

We next investigated whether UBE2G2 is involved in cell surface presentation of proinsulin-derived peptides. Mass spectrometry analysis of the HLA-A*02:01-eluted peptidome of UBE2G2 knockout cells revealed a drastic decrease in surface presentation of the proinsulin-derived B-chain autoantigen ($PPI_{B10-18}$) as compared to parental cells. Meanwhile, presentation of the proinsulin signal sequence-derived peptide ($PPI_{A15-24}$) and a panel of other peptides remained similar to parental cells (Fig 6E). Importantly, reintroduction of WT UBE2G2 increased $PPI_{B10-18}$ presentation while C89S UBE2G2 did only slightly, and presentation of other peptides was not drastically impacted as a function of UBE2G2 catalytic activity (Fig 6E). These results suggest that presentation of $PPI_{B10-18}$ is specifically altered as a function of UBE2G2 activity. Altogether, our data identified an important role for UBE2G2 in proinsulin degradation and presentation a proinsulin-derived B-chain autoantigen.

## Discussion

In this study, we tracked the origin of clinically important proinsulin B-chain peptides that are presented on the cell surface of pancreatic β-cells. We propose a model that involves dislocation of proinsulin from the ER into the cytosol, where it is degraded by the proteasome. In β-cells, this may occur if the ER is overloaded, as part of the unfolded protein response (UPR). B-cells produce large quantities of insulin whose biosynthesis requires cleavage of the PPI precursor and correct formation of multiple disulfide bonds. Any deviation from proper assembly results in accumulation of misfolded proinsulin, which triggers the UPR. As part of this response, ERAD-mediated dislocation of proinsulin will be initiated to relieve the ER from stress. At the heart of ERAD is ubiquitination of substrates by specific combinations of ubiquitin conjugating and ligating enzymes [18], and the work presented here suggests a role for the ERAD E3 enzyme HRD1 and the E2 enzyme UBE2G2 in proinsulin degradation. In the cytosol, dislocated glycoproteins lose their N-linked glycans by a cytosolic N-glycanase, which results in deamidation of the asparagine residue the glycan was attached to [19]. The proteasome degrades these substrates into peptides, which may return into the ER lumen to be loaded on HLA class I molecules. We made N-glycosylated mutant forms of proinsulin to demonstrate that presentation of a proinsulin B-chain peptide requires dislocation of proinsulin into the cytosol and involves UBE2G2. Presentation of the B-chain epitopes on HLA-A*02:01 and the cytotoxic function of the autoreactive CD8+ T cell population it evokes are of particular interest, since it has been shown that these factors play a major role during the early stages of T1D pathogenesis [3, 27].

Since ER-targeting of proinsulin was shown to drastically alter antigen presentation efficiency of proinsulin-derived peptides [25], identification of the intracellular origin of HLA class I-presented proinsulin epitopes is of high significance. For decades, this issue has been a

subject of intense debate. Proponents of the Defective Ribosomal Product (DRiP) hypothesis (reviewed in [28]) have claimed that HLA class I epitopes are mainly derived from immature, newly synthesized proteins or defective products that may have initiated from internal ribosomal entry sites, all of which are prone for proteasomal degradation. DRiPs stemming from secretory proteins may form either because of flawed translation onset in the cytoplasm or erroneous translocation into the ER. Opponents of the DRiP theory accredit normal turnover of fully functional proteins (also called 'retirees') as the primary source of peptides presented by HLA class I [29]. Here, we show that a B-chain proinsulin epitope originates from dislocated proinsulin species rather than from **1)** cytosolic degradation of a previously described pool of misfolded, and thus vulnerable, cytosolic PPI molecules translocated post-translationally in a Sec62-dependent fashion, **2)** proinsulin peptides that result from flawed translation onset in the cytoplasm, or **3)** a recently uncovered RTN3-dependent Erphagy pathway during which proinsulin remains unexposed to cytosolic N-glycanase [9, 17, 30]. Furthermore, since mature insulin molecules reside in β-cell secretory granules and are therefore unlikely to be targeted to ERAD for degradation, the PPI B-chain epitopes investigated here are unlikely to be a derivative of long-lived insulin molecules (i.e., insulin retirees). Thus, along the lines of the DRiP hypothesis, our data strongly suggest that proinsulin epitopes, such as $PPI_{B5-14}$ and $PPI_{B10-18}$, derive from immature proinsulin molecules that may have failed to achieve a proper conformation required for ER export and are consequently subjected to dislocation via ERAD.

Proper folding of proinsulin is mainly achieved through the formation of multiple disulfide bonds, which greatly enhanced its solubility and stability. Introducing a glycosylation motif by mutating a cysteine residue involved in disulfide bond formation likely affects protein stability and how it is selected for ERAD-mediated degradation. However, expression the C96N mutant does not qualitatively affect presentation of the PPI $B_{5-14}$ epitope, indicating that artificial introduction of a glycosylation site does not apprehend PI molecules to dislocate over the ER membrane and access cytosolic proteasomes. Our quantitative peptidome analysis revealed a general role for HRD1 in HLA-A*02:01-mediated antigen presentation in our model system through which it affects cell surface abundance of proinsulin-derived epitopes. In addition, it identified a specific role for UBE2G2 in antigen presentation of a clinically relevant proinsulin B-chain epitope ($PPI_{B10-18}$) in our model system.

Previous studies on factors involved in proinsulin degradation have focused on the misfolded C96Y proinsulin mutant expressed in Akita diabetic mice. This model has been used to study Mutant Insulin gene-induced Diabetes of Youth (MIDY). Here, CRISPR knockout has been used to validate the involvement of HRD1 in wild-type proinsulin degradation in K562 cells. Furthermore, the E2-specific CRISPR/Cas9 screen presented here identified UBE2G2, but not other ERAD-associated E2's UBE2J1 or UBE2J2, as an essential E2 ubiquitin conjugating enzyme responsible for proinsulin degradation. Although HRD1 has been shown to join forces with UBE2J1 [20], our data now suggest that HRD1 and UBE2G2 can also cooperate in mammalian ERAD. This is in line with the model in which UBE2G2 is responsible for elongation of ubiquitin chains on substrates that are primed by UBE2J1 family member UBC6 but can also independently drive poly-ubiquitination on ERAD substrates [31]. Since ubiquitination of proinsulin is thought to occur on K88 [32], it would be interesting to investigate whether the HRD1 and UBE2G2 target proinsulin for modification on K88 to induce its degradation by the proteasome.

Since β-cells synthesize enormous quantities of proinsulin molecules, it is not surprising that a substantial proportion of molecules encounters problems in attaining a mature conformation and is selected for degradation via ERAD [33]. This cascade may eventually result in presentation of proinsulin-derived B-chain autoantigens on the β-cell surface. While the ER of K562 cells may not reflect the natural proinsulin folding environment, these cells display

relevant PPI-derived epitopes on HLA-A*02:01 [12] and likely resemble a genuine β-cell regarding the pathways involved in generation of these autoantigens. Nevertheless, it would be interesting to validate these results in a more physiological setting. Along these lines, future experiments could uncover or pharmacologically inhibit the activity HRD1 and UBE2G2 in islets derived from T1D patients versus healthy individuals.

In a subset of T1D patients, proinsulin has been found to accumulate in the ER [34]. Accumulation of incorrectly folded proinsulin leads to ER stress, and this has been shown to play a significant role in the production of neo- or auto-antigens, β-cell destruction and (the onset of) T1D (reviewed in [35, 36]). Correct folding of proinsulin and its exit from the ER mainly depend on formation of at least two out of three conserved disulfide bonds, and the ER of β-cells harbors specialized enzymes including PDI and ERO1β to catalyze this process [10, 35]. Modulation of the abundance of these proteins has been shown to enhance proinsulin secretion and relieve the β-cell from ER stress [15, 37]. Next to these redox enzymes, factors such as SDF2L1 [14] and Grp170 [17, 38, 39] are implicated in chaperone complexes that aid in proinsulin folding, shifting proinsulin homeostasis towards export versus accumulation and degradation. Our data suggests that presentation of proinsulin B-chain autoantigens depends on ERAD-mediated dislocation and may thus rely on proinsulin degradation by the HRD1/UBE2G2 ubiquitination complex. In this light, reducing the proinsulin degradation load on ERAD by improving the β-cell's oxidative folding environment may decrease the number of self-peptides available in the prediabetic pancreas for presentation to autoreactive CD8[+] T cells. Since the UBE2G2-dependent B-chain antigen (PPI$_{B10-18}$) has been implicated in graft rejection after islet transplantation [27], intervening with its cell surface presentation, for instance via genetic modification [40], may improve graft acceptance. An approach as mentioned above may concurrently lower ER stress, improve proinsulin maturation and export, and decrease ERAD-mediated presentation of proinsulin autoantigens in β-cells. Such a tactic would hit three birds with one stone in combatting T1D.

## Supporting information

**S1 Fig. PPI(B5-14) peptide variants identified by mass spectrometry of mutant proinsulin expressing HLA-peptidomes.** (A-C) (Top) Shown are mass spectrometry spectra of WT (A) or mutant (B, C) proinsulin-derived peptides, indicating the identification of proinsulin peptides as eluted from (mutant) proinsulin expressing cells shown in Figs 5 and 6. (Bottom) spectra show measurements of corresponding peptides for validation.
(TIF)

**S2 Fig. Validation of the role of HRD1 in proinsulin degradation in two additional HRD1 KO clones. A.** K562 cells stably expressing HLA-A*02:01 and PPI were transduced with three different CRISPR-Cas9 vectors containing a gRNA sequence directed against the N-terminal region of the HRD1 gene and selected with puromycin. Polyclonal cell populations were analyzed for HRD1 and proinsulin levels by Western blotting. **B.** Monoclonal knockout cell lines were generated by limiting dilution of gRNA#3 polyclonal cells shown in (A). Genomic DNA was isolated and sequenced for the presence of deletions within the gRNA target region. Both HRD1 alleles of clones 2 and 3 were aligned to a reference sequence (NCBI gene entry 84447). **C.** K562 cells from (B) were retransduced with an empty cDNA vector (KO) or a cDNA vector encoding HRD1 (WT), or a catalytically inactive mutant (C1A). Cells were sorted on mAmetrin expression to obtain a pure population. Next cell lysates were analyzed by WB for expression of HRD1 and proinsulin. Human transferrin receptor was used as a loading control.
(TIF)

**S3 Fig. Validation of the role of UBE2G2 in proinsulin degradation in two additional UBE2G2 KO clones. A.** K562 cells stably expressing HLA-A*02:01 and PPI were transduced with three different CRISPR-Cas9 vectors containing a gRNA sequence directed against the N-terminal region of the HRD1 gene and selected with puromycin. Polyclonal cell populations were analyzed for HRD1 and proinsulin levels by Western blotting. **B.** Monoclonal knockout cell lines were generated by limiting dilution of gRNA#1 polyclonal cells shown in (A). Genomic DNA was isolated and sequenced for the presence of deletions within the gRNA target region. Both HRD1 alleles of clones 2 and 3 were aligned to a reference sequence (NCBI gene entry 7327). **C.** K562 cells from (B) were retransduced with an empty cDNA vector (KO) or a cDNA vector encoding UBE2G2 (WT), or a catalytically inactive mutant (C89S). Cells were sorted on mAmetrin expression to obtain a pure population. Next cell lysates were analyzed by WB for expression of HRD1 and proinsulin. Human transferrin receptor was used as a loading control.
(TIF)

**S4 Fig. Raw and uncropped versions of Western blot scans from main figures.**
(TIF)

**S1 Table. List of gRNA sequences used in arrayed CRISPR-Cas9-mediated screen.**
(TIF)

**S2 Table. Numerical values underlying graphs shown in this study.**
(XLSX)

**S3 Table. HLA-eluted peptidome analysis in (rescued) HRD1 and UBE2G2 K.O. cells.**
(XLSX)

**S1 Graphical abstract.**
(TIF)

## Acknowledgments

We would like to thank dr. Ilana Berlin (LUMC) and dr. A. Zaldumbide (LUMC) for critical reading of the manuscript.

## Author Contributions

**Conceptualization:** Hanneke Hoelen, Emmanuel J. H. J. Wiertz.

**Data curation:** Tom Cremer, Hanneke Hoelen, Peter A. van Veelen.

**Formal analysis:** Tom Cremer, Hanneke Hoelen, George M. Janssen.

**Funding acquisition:** Emmanuel J. H. J. Wiertz.

**Investigation:** Tom Cremer, Hanneke Hoelen, George M. Janssen.

**Methodology:** Hanneke Hoelen, Michael L. van de Weijer, George M. Janssen, Peter A. van Veelen.

**Project administration:** Hanneke Hoelen, Ana I. Costa.

**Resources:** Michael L. van de Weijer, Robert Jan Lebbink.

**Supervision:** Hanneke Hoelen, Peter A. van Veelen, Robert Jan Lebbink, Emmanuel J. H. J. Wiertz.

**Visualization:** Tom Cremer.

**Writing – original draft:** Tom Cremer, Emmanuel J. H. J. Wiertz.

**Writing – review & editing:** Ana I. Costa.

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
