## [Decision Letter · Decision Letter 0]

13 Jul 2023

PONE-D-23-18345Proinsulin degradation and presentation of a proinsulin B-chain autoantigen involves ER-associated protein degradation (ERAD)-enzyme UBE2G2PLOS ONE

Dear Dr. Cremer,

Thank you for submitting your manuscript to PLOS ONE. After careful consideration, we feel that it has merit but does not fully meet PLOS ONE’s publication criteria as it currently stands. Therefore, we invite you to submit a revised version of the manuscript that addresses the points raised during the review process.

The reviewers and myself agree that the findings are potentially of importance, but both reviewers raised very important points that need to be addressed.In additions to their comments, I'm concerned that the low n values in some of the experiments (n=3 in Figure 2D and 4D - both of which failed to be referenced in the Results section as well) are going to make statistical significance extremely challenging unless additional repeats are provided. And while convincing, the blots in Figure 5 also lack quantification of any sort.==============================

We look forward to receiving your revised manuscript.

Kind regards,

Corentin Cras-Méneur, Ph.D.

Academic Editor

PLOS ONE

Journal Requirements:

6. Please include a copy of Table 1 which you refer to in your text on page 6.

Reviewers' comments:

Reviewer's Responses to Questions

**Comments to the Author**

1. Is the manuscript technically sound, and do the data support the conclusions?

Reviewer #1: Partly

Reviewer #2: Partly

2. Has the statistical analysis been performed appropriately and rigorously? 

Reviewer #1: No

Reviewer #2: N/A

3. Have the authors made all data underlying the findings in their manuscript fully available?

Reviewer #1: No

Reviewer #2: Yes

4. Is the manuscript presented in an intelligible fashion and written in standard English?

Reviewer #1: Yes

Reviewer #2: Yes

5. Review Comments to the Author

Reviewer #1: In this manuscript, the authors seek to better understand how proinsulin is degraded by ER-associated degradation (ERAD) and whether this process contributes to generation of (potentially abnormal) peptides that are then used by HLA in antigen presentation. Understanding this process is of clinical importance, as presentation of (abnormal) proinsulin peptides has been proposed as a trigger for autoimmune diabetes. Although the data are interesting and shown with a reasonable experimental approach, there are a few outstanding issues needed to confirm the findings summarized in the proposed model.

Major comments:

• No description/summary of statistical analyses; for graphs only averages +/- SD is shown (not actual values)

• In Figures 2 and 4, it would be helpful to show the response of true control (“EV”) cells alongside the knockout and re-transduced cells. Do the knockout + HRD1/2G2 WT cells treated with cycloheximide mimic what happens in treated cells with endogenous HRD1/2G2? Or does overexpression of the replaced HRD1/2G2 itself affect proinsulin turnover?

• Were the (N-glycosylated mutant) proinsulin forms transfected into HRD1 KO cells? This is an important experimental control to show that HRD1-mediated ERAD itself is necessary to generate the proinsulin peptides that are later presented by HLA.

• There was little introduction to the role of TAP in the antigen presentation process here. Is TAP required for ER uptake and attachment of substrates to the HLA molecules that are presenting the proinsulin peptides, or are there other mechanisms? It would be helpful to show that TAP knockout abolishes uptake and association of the mutant proinsulin peptides with HLA, as proposed in Figure 1.

Minor comments:

• On line 200, text should refer to Figure 2D, not 1D

• For Figure 2D, would recommend using similar labeling as Figure 4D – for example, I believe the authors are referring to “HRD1 KO” cells instead of “EV” as shown

• Frequent switching between “(pre)proinsulin” and "PPI”/”PI” makes the text difficult to read at times

Reviewer #2: In the manuscript, PONE-D-23-18345: Proinsulin degradation and presentation of a proinsulin B-chain autoantigen involves ER-associated protein degradation (ERAD)-enzyme UBE2G2, Cremer et al report UBE2G2 as a plausible Ubiquitin-conjugating enzyme (E2) that is involved in proinsulin degradation, and that the degraded products (peptides) could serve as autoantigens, especially in the context of type-1 diabetes (T1D). The experimental design, methods, and interpretation of the results are good, and the discussion is well-written. The idea of preproinsulin being a primary degradation substrate is interesting, and the results relating to antigen presentation certainly have clinical value. While K562 cells appear to be a decent model to explore intracellular events, given the premise of the study is proinsulin degradation, the reviewer strongly recommends the usage of pancreatic beta cells (cultured cell lines like INS1 or MIN6) in this study. Specific comments for the authors can be found below.

1. The authors should determine whether UBE2G2 displays the suggested function in pancreatic beta cells. Would UBE2G2 knockout or siRNA-based knockdown of endogenous UBE2G2 or similar proteins in beta cells produce a phenotype wherein the undegraded substrate (proinsulin) accumulates in cells? Please include suitable experiments to address this question. The authors may consider using wild-type or MIDY mutants of proinsulin for this study.

2. As the authors describe in the introduction, proinsulin after synthesis enters the endoplasmic reticulum (ER), gets folded, and is trafficked to Golgi and secretory granules, making insulin en route. Proinsulin that does not proceed from ER to Golgi is presumably degraded by ERAD after a retrotranslocation via the ERAD channel. Considering these are the events that are known to occur naturally, how do the authors justify the experiments, wherein the glycosylation sites are artificially introduced, possibly enhancing the half-life of preproinsulin in cells? Though previous studies using beta cells indicate that preproinsulin may serve as autoantigens, the current model and the approach used by the authors in the current work seems impracticable. Once again, the reviewer emphasizes on the utilization of beta cells.

3. The authors state that the newly-synthesized proinsulin is misfolded in K562 cells. The reviewer suggests this assertion must be substantiated with experiments. It should also be addressed whether substituting proinsulin Cys residues with other amino acids itself may drive misfolding of the protein.

4. Include a brief discussion of post-translational modifications proinsulin may undergo in the beta cells, and how these may stabilize (or destabilize) preproinsulin making it a viable degradation substrate and/or autoantigen.

5. Optional experiment: If it is possible, please show the levels or activity of UBE2G2 (or similar related protein(s) in beta cells) in purified pancreatic islets from T1D models (NOD or Akita) versus healthy controls. This would greatly enhance the physiological viewpoint of the manuscript.

6. PLOS authors have the option to publish the peer review history of their article (what does this mean?). If published, this will include your full peer review and any attached files.

Reviewer #1: No

Reviewer #2: No

---

## [Author Response · Author response to Decision Letter 0]

3 Nov 2023

Point-by-point response to reviewers:

Reviewer #1: In this manuscript, the authors seek to better understand how proinsulin is degraded by ER-associated degradation (ERAD) and whether this process contributes to generation of (potentially abnormal) peptides that are then used by HLA in antigen presentation. Understanding this process is of clinical importance, as presentation of (abnormal) proinsulin peptides has been proposed as a trigger for autoimmune diabetes. Although the data are interesting and shown with a reasonable experimental approach, there are a few outstanding issues needed to confirm the findings summarized in the proposed model.

We thank the Reviewer for her/his appreciation of our experimental data and interest in our study; we also appreciate the constructive suggestions to improve the manuscript. 

Major comments: 

• No description/summary of statistical analyses; for graphs only averages +/- SD is shown (not actual values) 

We sincerely apologize for omitting the summary of statistics and have provided a detailed analysis in the revised figures with appropriate descriptions in the figure legends. We decided not to show all replicate values because that would severely decrease comprehensibility of the graph but the numerical data points values are now shown as supplementary information. We hope to have addressed this issue sufficiently. 

• In Figures 2 and 4, it would be helpful to show the response of true control (“EV”) cells alongside the knockout and re-transduced cells. Do the knockout + HRD1/2G2 WT cells treated with cycloheximide mimic what happens in treated cells with endogenous HRD1/2G2? Or does overexpression of the replaced HRD1/2G2 itself affect proinsulin turnover? 

We agree with the reviewer that ectopic overexpression of enzymes may alter downstream degradation kinetics. Considering UBE2G2, protein levels during ectopic expression were similar to those observed in the context of endogenous expression (Fig. 4C). Ectopic expression does not appreciably alter PI steady-state levels. 

For HRD1, it should be taken into consideration that E3 ligase expression is often a rate-limiting step. Therefore, overexpression is expected to affect PI steady-state levels and degradation kinetics, which was previously shown for other HRD1 substrates [1]. However, the primary aim of our CHX chase experiments was to assess the effect of E2 or E3 enzymatic activity on proinsulin degradation kinetics within the same clonal setting, and therefore have excluded the heterogeneous parental population from these experiments. Therefore, we reconstituted the WT or catalytically inactive enzymes to comparable levels, which allows specific measurement of the influence of HRD1 and UBE2G2 catalytic residues on proinsulin degradation and excludes the influence of potential overexpression effects on this conclusion. We thank the reviewer for this comment and hope to have addressed it sufficiently.

• Were the (N-glycosylated mutant) proinsulin forms transfected into HRD1 KO cells? This is an important experimental control to show that HRD1-mediated ERAD itself is necessary to generate the proinsulin peptides that are later presented by HLA.

We thank the reviewer for suggesting this important control experiment. As a follow-up experiment, we performed mass spectrometry on the eluted HLA peptidome of our set of (reconstituted) HRD1 KO cell lines and manually quantified the peak height, comparing them to control parental K562-A2*01 cells. In HRD1 KO cells or cells expressing the catalytically inactive HRD1 protein, we observed a dramatic decrease in the number of peptides derived from the proinsulin B-chain. However, we also observed a decrease in cell surface presented peptides as a function of HRD1 activity (see Fig. R1). Thus, while HRD1 activity seems to be important for proinsulin degradation, it also seems to affect HLA*A2-dependent antigen presentation in general. Although interesting, these data describing a general effect of HRD1 ablation on antigen presentation is out of the scope of our manuscript and therefore we decided not to include these in our revised version. 

We acknowledge that we cannot directly link HRD1 and UBE2G2 activity to presentation of proinsulin-derived peptides although we point out that this requires dislocation over the ER membrane via ERAD, of which HRD1 and UBE2G2 are core enzymes. 

Figure R1: presentation of a proinsulin B-chain peptide and other HLA-A2*01 bound peptides require HRD1 catalytic activity. Eluted peptide intensities were manually quantified (peak height) and normalized to parental K562-A2-PPI cells (control)

• There was little introduction to the role of TAP in the antigen presentation process here. Is TAP required for ER uptake and attachment of substrates to the HLA molecules that are presenting the proinsulin peptides, or are there other mechanisms? It would be helpful to show that TAP knockout abolishes uptake and association of the mutant proinsulin peptides with HLA, as proposed in Figure 1. 

We thank the reviewer for pointing out this confusing issue. Indeed, TAP-deficient individuals do not completely lack CD8-responses, pointing at TAP-independent mechanisms of peptide import into the ER [2]. Also, proinsulin-derived peptides such as signal-sequence-derived peptides [3] can be presented by TAP-independent mechanisms. These nuances have been addressed in our revised manuscript (lines 87-88). Although we agree it would be interesting to verify the role of TAP in the loading of proinsulin antigens on HLA class I, our study focuses on processes that occur prior to ER import of peptides. Therefore, we believe that experiments on the role of TAP in proinsulin presentation are out of the scope of this manuscript. 

Minor comments: 

• On line 200, text should refer to Figure 2D, not 1D 

• For Figure 2D, would recommend using similar labeling as Figure 4D – for example, I believe the authors are referring to “HRD1 KO” cells instead of “EV” as shown

We thank the reviewer for these minor comments and have addressed these in the revised manuscript.

• Frequent switching between “(pre)proinsulin” and "PPI”/”PI” makes the text difficult to read at times.

We agree that nomenclature on proinsulin can be confusing at times. For clarity, we use preproinsulin (PPI) in the context of DNA/RNA and peptides derived from the unprocessed form of insulin, while PI is used for proinsulin molecules from which the signal peptide has been removed after entering the ER. We have clarified the use of these abbreviations in our revised introduction. 

Reviewer #2: 

In the manuscript, PONE-D-23-18345: Proinsulin degradation and presentation of a proinsulin B-chain autoantigen involves ER-associated protein degradation (ERAD)-enzyme UBE2G2, Cremer et al report UBE2G2 as a plausible Ubiquitin-conjugating enzyme (E2) that is involved in proinsulin degradation, and that the degraded products (peptides) could serve as autoantigens, especially in the context of type-1 diabetes (T1D). The experimental design, methods, and interpretation of the results are good, and the discussion is well-written. The idea of preproinsulin being a primary degradation substrate is interesting, and the results relating to antigen presentation certainly have clinical value. While K562 cells appear to be a decent model to explore intracellular events, given the premise of the study is proinsulin degradation, the reviewer strongly recommends the usage of pancreatic beta cells (cultured cell lines like INS1 or MIN6) in this study. Specific comments for the authors can be found below. 

We thank the Reviewer for her/his favorable assessment of our study and appreciate the constructive suggestions offered to improve it. We agree that expanding the results obtained in our K562 model system to the pancreatic beta cell lines recommended by the reviewer would greatly extend the clinical value of our study. However, this would also increase the complexity of the studies. We have explored the use of pancreatic beta cell lines and have run into many practical issues. We mentioned these suggestions in our Discussion section and hope that we have sufficiently addressed the limitations of our model system in our revised manuscript.

1. The authors should determine whether UBE2G2 displays the suggested function in

pancreatic beta cells. Would UBE2G2 knockout or siRNA-based knockdown of endogenous

UBE2G2 or similar proteins in beta cells produce a phenotype wherein the undegraded

substrate (proinsulin) accumulates in cells? Please include suitable experiments to address this question. The authors may consider using wild-type or MIDY mutants of proinsulin for this study.

We thank the reviewer for her/his suggestion. As ERAD mechanisms are conserved across species and between cell types, especially regarding soluble substrates, we believe that our findings on the role of core ERAD machinery in degradation of proinsulin can be extended to other cell types, such as pancreatic beta cells. We have addressed this suggestion in our revised Discussion section (lines 316-317). 

2. As the authors describe in the introduction, proinsulin after synthesis enters the endoplasmic reticulum (ER), gets folded, and is trafficked to Golgi and secretory granules, making insulin en route. Proinsulin that does not proceed from ER to Golgi is presumably degraded by ERAD after a retrotranslocation via the ERAD channel. Considering these are the events that are known to occur naturally, how do the authors justify the experiments, wherein the glycosylation sites are artificially introduced, possibly enhancing the half-life of preproinsulin in cells? Though previous studies using beta cells indicate that preproinsulin may serve as autoantigens, the current model and the approach used by the authors in the current work seems impracticable. Once again, the reviewer emphasizes on the utilization of beta cells.

We agree with the reviewer that artificially introducing mutations in substrate proteins may steer away substrates from their natural degradation pathways. Mutation of cysteine residues involved in disulfide bond formation, like C31, would indeed preclude the successful folding of proinsulin molecules and affect their stability. However, cells expressing our C96N mutant also present the natural B5-14 epitope, indicating that it does not affect the dislocation of the protein as it still accesses cytosolic proteasomes that generate the same B5-14 epitope. We therefore believe that it is safe to conclude that the biochemical pathways that generate this specific antigen are still used as natural, i.e. dislocation over the ER membrane into the cytosol and peptide generation by the proteasome. In our revised manuscript, we specify our claims only regarding this this antigen specifically.

3. The authors state that the newly-synthesized proinsulin is misfolded in K562 cells. The

reviewer suggests this assertion must be substantiated with experiments. It should also be

addressed whether substituting proinsulin Cys residues with other amino acids itself may drive misfolding of the protein.

We thank the reviewer for her/his feedback on our Discussion section. We agree that the substitution of proinsulin Cys molecules likely affects protein folding and stability, but the results from our C96N mutant suggest that this does not qualitatively alter the presentation of the PPI B5-14 epitope. We have discussed this in more detail in our revised manuscript (lines 296-298). We have also removed the statement about misfolding of WT proinsulin in our model system as we do not provide data that proves this. 

4. Include a brief discussion of post-translational modifications proinsulin may undergo in the

beta cells, and how these may stabilize (or destabilize) preproinsulin making it a viable

degradation substrate and/or autoantigen.

We appreciate the suggestion to discussthe details of proinsulin stability through post-translational modification and now mention the specifics of proinsulin ubiquitination in our revised manuscript (lines 307-309). Furthermore, we have expanded our discussion on proinsulin stability through disulfide bond formation and how our glycosylated mutants may affect this (lines 293-295). 

5. Optional experiment: If it is possible, please show the levels or activity of UBE2G2 (or similar

related protein(s) in beta cells) in purified pancreatic islets from T1D models (NOD or Akita)

versus healthy controls. This would greatly enhance the physiological viewpoint of the

manuscript.

We thank the reviewer for their excellent suggestions for extra experiments. Expanding our findings in beta cells would indeed greatly enhance the study from a physiological viewpoint. We have added the suggestion for this experiment in our revised Discussion section (lines 317-319). 

References

1. Liu L, Long H, Wu Y, Li H, Dong L, Zhong JL, et al. HRD1-mediated PTEN degradation promotes cell proliferation and hepatocellular carcinoma progression. Cell Signal. 2018;50:90-9. doi: 10.1016/j.cellsig.2018.06.011. PubMed PMID: 29958993.

2. De la Salle H, Saulquin X, Mansour I, Klayme S, Fricker D, Zimmer J, et al. Asymptomatic deficiency in the peptide transporter associated to antigen processing (TAP). Clinical & Experimental Immunology. 2002;128(3):525-31.

3. Kronenberg-Versteeg D, Eichmann M, Russell MA, de Ru A, Hehn B, Yusuf N, et al. Molecular Pathways for Immune Recognition of Preproinsulin Signal Peptide in Type 1 Diabetes. Diabetes. 2018;67(4):687-96. doi: 10.2337/db17-0021. PubMed PMID: 29343547.

---

## [Decision Letter · Decision Letter 1]

28 Nov 2023

PONE-D-23-18345R1Proinsulin degradation and presentation of a proinsulin B-chain autoantigen involves ER-associated protein degradation (ERAD)-enzyme UBE2G2PLOS ONE

Dear Dr. Cremer,

Thank you for submitting your manuscript to PLOS ONE. After careful consideration, we feel that it has merit but does not fully meet PLOS ONE’s publication criteria as it currently stands. Therefore, we invite you to submit a revised version of the manuscript that addresses the points raised during the review process.

Both reviewers requested a major revision for the manuscript.While most of the comments of one of the reviewers was addressed, none of the experiments suggested by the other one have been performed. Both reviewers also mentioned that the conclusions were overreaching.

We look forward to receiving your revised manuscript.

Kind regards,

Corentin Cras-Méneur, Ph.D.

Academic Editor

PLOS ONE

Additional Editor Comments:

Both reviewers requested a major revision for the manuscript. While most of the comments of one of the reviewers was addressed, none of the experiments suggested by the other one have been performed. Both reviewers also mentioned that the conclusions were overreaching.

Reviewers' comments:

Reviewer's Responses to Questions

**Comments to the Author**

1. If the authors have adequately addressed your comments raised in a previous round of review and you feel that this manuscript is now acceptable for publication, you may indicate that here to bypass the “Comments to the Author” section, enter your conflict of interest statement in the “Confidential to Editor” section, and submit your "Accept" recommendation.

Reviewer #1: (No Response)

Reviewer #2: (No Response)

2. Is the manuscript technically sound, and do the data support the conclusions?

Reviewer #1: Partly

Reviewer #2: No

3. Has the statistical analysis been performed appropriately and rigorously? 

Reviewer #1: Yes

Reviewer #2: Yes

4. Have the authors made all data underlying the findings in their manuscript fully available?

Reviewer #1: Yes

Reviewer #2: Yes

5. Is the manuscript presented in an intelligible fashion and written in standard English?

Reviewer #1: Yes

Reviewer #2: Yes

6. Review Comments to the Author

Reviewer #1: Most of my concerns have been sufficiently addressed. However, I believe the manuscript’s conclusion is still stated too strongly based on the data presented. I appreciate the authors’ attempt to show that generation of proinsulin-derived peptides in HRD1 KO cells is greatly reduced compared to controls (Figure R1), and understand their interpretation that the concomitant reduction in antigen presentation capacity in the HRD1 KO cells confounds the conclusion that HRD1 inactivation directly affects proinsulin-derived peptide antigen presentation. As stated in the response, “We acknowledge that we cannot directly link HRD1 and UBE2G2 activity to presentation of proinsulin-derived peptides although we point out that this requires dislocation over the ER membrane via ERAD, of which HRD1 and UBE2G2 are core enzymes.” However, this leaves little data presented to link the role of HRD1 ERAD (with UBE2G2) in stabilizing wild-type proinsulin to the observed findings that mutated proinsulin forms appear to be shuttled to the cytoplasm, where they are modified before antigen presentation. Since using HRD1 KO cells in the experiments shown in Figure 6 may lead to confounded results, the experiment could be repeated with UBE2G2 KO cells – and according to the proposed model, there should be a reduction in modified epitopes detected compared to using cells with functional UBE2G2-HRD1 ERAD. If this experiment cannot be accomplished, the conclusions should be modified to deemphasize the direct link as proposed in the graphical abstract and throughout the Discussion section (for example, lines 350-352, “Our data strongly suggests that degradation of proinsulin through HRD1 and UBE2G2-mediated ERAD results in presentation of a proinsulin B-chain autoantigens.”). Alternatively, adding Figure R1 to the manuscript with a brief discussion of the conclusions from that experiment would be helpful to explain why the direct experiment is not possible.

Reviewer #2: No experiments suggested by the reviewer have been performed. While the authors intend to do additional relevant work in the future, the reviewer again emphasizes that the UBE2G2 function in the pancreatic beta cells should be determined as part of this manuscript. Does UBE2G2 knockdown, knockout, or functional inhibition (using CW3 compound, for example) in the beta cells lead to the accumulation of undegraded proinsulin? Please note that the reviewer is not insisting on using purified islets from diabetes models but suggesting experiments using beta cell lines (MIN6 or INS1). Without these studies, the data in the manuscript does not satisfactorily support the conclusions.

7. PLOS authors have the option to publish the peer review history of their article (what does this mean?). If published, this will include your full peer review and any attached files.

Reviewer #1: No

Reviewer #2: No

---

## [Author Response · Author response to Decision Letter 1]

29 Feb 2024

Reviewer #1: Most of my concerns have been sufficiently addressed. However, I believe the manuscript’s conclusion is still stated too strongly based on the data presented. I appreciate the authors’ attempt to show that generation of proinsulin-derived peptides in HRD1 KO cells is greatly reduced compared to controls (Figure R1), and understand their interpretation that the concomitant reduction in antigen presentation capacity in the HRD1 KO cells confounds the conclusion that HRD1 inactivation directly affects proinsulin-derived peptide antigen presentation. As stated in the response, “We acknowledge that we cannot directly link HRD1 and UBE2G2 activity to presentation of proinsulin-derived peptides although we point out that this requires dislocation over the ER membrane via ERAD, of which HRD1 and UBE2G2 are core enzymes.” However, this leaves little data presented to link the role of HRD1 ERAD (with UBE2G2) in stabilizing wild-type proinsulin to the observed findings that mutated proinsulin forms appear to be shuttled to the cytoplasm, where they are modified before antigen presentation. Since using HRD1 KO cells in the experiments shown in Figure 6 may lead to confounded results, the experiment could be repeated with UBE2G2 KO cells – and according to the proposed model, there should be a reduction in modified epitopes detected compared to using cells with functional UBE2G2-HRD1 ERAD. If this experiment cannot be accomplished, the conclusions should be modified to deemphasize the direct link as proposed in the graphical abstract and throughout the Discussion section (for example, lines 350-352, “Our data strongly suggests that degradation of proinsulin through HRD1 and UBE2G2-mediated ERAD results in presentation of a proinsulin B-chain autoantigens.”). Alternatively, adding Figure R1 to the manuscript with a brief discussion of the conclusions from that experiment would be helpful to explain why the direct experiment is not possible.

We thank the reviewer for his suggestion to include the quantitative peptidome analysis in our manuscript. Previously, we have also performed the experiment in the (rescued) UBE2G2 KO cell panel, and are pleased to report its reanalysis upon request of the reviewer. The data in Fig. 6E suggests a specific role for this enzyme in the presentation of a PPI-derived B-chain autoantigen, while the previously communicated data in Fig. 4E (HRD1 KO) suggests a general role for HRD1 in antigen presentation in our model system. In line with our model, these data link UBE2G2 to proinsulin antigen presentation. We believe that the full dataset (including the HRD1 KO series) is of great value to the scientific community and will share it as a supplemental table in our manuscript. All spectra are uploaded to the PRIDE platform.

Reviewer #2: No experiments suggested by the reviewer have been performed. While the authors intend to do additional relevant work in the future, the reviewer again emphasizes that the UBE2G2 function in the pancreatic beta cells should be determined as part of this manuscript. Does UBE2G2 knockdown, knockout, or functional inhibition (using CW3 compound, for example) in the beta cells lead to the accumulation of undegraded proinsulin? Please note that the reviewer is not insisting on using purified islets from diabetes models but suggesting experiments using beta cell lines (MIN6 or INS1). Without these studies, the data in the manuscript does not satisfactorily support the conclusions.

We thank the reviewer for his/her suggestions to improve our manuscript by performing additional experiments in mouse beta cell lines. However, since the lab has been discontinued and all persons involved have been involved in different projects for some time now, we are not able to set up these experiments. Nevertheless, we have discussed the value of these experiments in our discussion section and have revised the relevant conclusions throughout the text, and emphasized the use of our K562 model in our graphical abstract.

---

## [Decision Letter · Decision Letter 2]

15 Mar 2024

Proinsulin degradation and presentation of a proinsulin B-chain autoantigen involves ER-associated protein degradation (ERAD)-enzyme UBE2G2

PONE-D-23-18345R2

Dear Dr. Cremer,

We’re pleased to inform you that your manuscript has been judged scientifically suitable for publication and will be formally accepted for publication once it meets all outstanding technical requirements.

Kind regards,

Corentin Cras-Méneur, Ph.D.

Academic Editor

PLOS ONE

Additional Editor Comments (optional):

All major comments have been addressed in the revised manuscript. Reviewer #1 requested some minor modifications that could be beneficial.

Reviewers' comments:

Reviewer's Responses to Questions

**Comments to the Author**

1. If the authors have adequately addressed your comments raised in a previous round of review and you feel that this manuscript is now acceptable for publication, you may indicate that here to bypass the “Comments to the Author” section, enter your conflict of interest statement in the “Confidential to Editor” section, and submit your "Accept" recommendation.

Reviewer #1: All comments have been addressed

Reviewer #2: All comments have been addressed

2. Is the manuscript technically sound, and do the data support the conclusions?

Reviewer #1: Yes

Reviewer #2: Yes

3. Has the statistical analysis been performed appropriately and rigorously? 

Reviewer #1: Yes

Reviewer #2: Yes

4. Have the authors made all data underlying the findings in their manuscript fully available?

Reviewer #1: Yes

Reviewer #2: Yes

5. Is the manuscript presented in an intelligible fashion and written in standard English?

Reviewer #1: Yes

Reviewer #2: Yes

6. Review Comments to the Author

Reviewer #1: The additional data and text modification in this version seem sufficient for publication. I would suggest that "TAP" be removed from Figure 1 as this was not directly tested in the presented studies - or at least modifying the figure legend to say that "epitopes are imported into the ER by transporters such as TAP...".

Reviewer #2: (No Response)

7. PLOS authors have the option to publish the peer review history of their article (what does this mean?). If published, this will include your full peer review and any attached files.

Reviewer #1: No

Reviewer #2: No

---

## [Editor Report · Acceptance letter]

30 Apr 2024

PONE-D-23-18345R2 

PLOS ONE

Dear Dr. Cremer, 

I'm pleased to inform you that your manuscript has been deemed suitable for publication in PLOS ONE. Congratulations! Your manuscript is now being handed over to our production team.

Kind regards, 

on behalf of

Dr. Corentin Cras-Méneur 

Academic Editor

PLOS ONE